# Quantitative Risk Assessment for the Introduction of Carbapenem-Resistant Enterobacteriaceae (CPE) into Dutch Livestock Farms

**DOI:** 10.3390/antibiotics11020281

**Published:** 2022-02-21

**Authors:** Natcha Dankittipong, Egil A. J. Fischer, Manon Swanenburg, Jaap A. Wagenaar, Arjan J. Stegeman, Clazien J. de Vos

**Affiliations:** 1Department Population Health Sciences, Farm Animal Health, Utrecht University, Martinus G. de Bruingebouw, Yalelaan 7, 3584 CL Utrecht, The Netherlands; e.a.j.fischer@uu.nl (E.A.J.F.); j.a.stegeman@uu.nl (A.J.S.); 2Wageningen Bioveterinary Research, Wageningen University & Research, Houtribweg 39, 8221 RA Lelystad, The Netherlands; manon.swanenburg@wur.nl (M.S.); clazien.devos@wur.nl (C.J.d.V.); 3Department Biomolecular Health Science, Infectious Diseases & Immunology, Utrecht University, Androclusgebouw, Yalelaan 1, 3584 CL Utrecht, The Netherlands; j.wagenaar@uu.nl

**Keywords:** carbapenems, CPE, meat-producing animal, companion animal, travelers, feed, risk assessment, introduction risk, stochastic risk model

## Abstract

Early detection of emerging carbapenem-resistant Enterobacteriaceae (CPE) in food-producing animals is essential to control the spread of CPE. We assessed the risk of CPE introduction from imported livestock, livestock feed, companion animals, hospital patients, and returning travelers into livestock farms in The Netherlands, including (1) broiler, (2) broiler breeder, (3) fattening pig, (4) breeding pig, (5) farrow-to-finish pig, and (6) veal calf farms. The expected annual number of introductions was calculated from the number of farms exposed to each CPE source and the probability that at least one animal in an exposed farm is colonized. The total number of farms with CPE colonization was estimated to be the highest for fattening pig farms, whereas the probability of introduction for an individual farm was the highest for broiler farms. Livestock feed and imported livestock are the most likely sources of CPE introduction into Dutch livestock farms. Sensitivity analysis indicated that the number of fattening pig farms determined the number of high introductions in fattening pigs from feed, and that uncertainty on CPE prevalence impacted the absolute risk estimate for all farm types. The results of this study can be used to inform risk-based surveillance for CPE in livestock farms.

## 1. Introduction

Antimicrobial-resistant (AMR) bacteria have been one of the greatest public health challenges since the 1950s [1]. Increased use of broad-spectrum antibiotics has resulted in a race between resistant bacteria and treatments. The lagging development of new antibiotics and the speed at which resistance emerges are propelling the healthcare sector toward using “drugs of last resort”, administered only after other antibiotics have failed. One antimicrobial class of last resort, carbapenems, represents extremely potent, broad-spectrum drugs for treating serious infections, primarily from multidrug-resistant Enterobacteriaceae [2]. Enterobacteriaceae with carbapenem-resistant genes have a 50% mortality rate in humans due to the absence of alternative antibiotic treatments [3]. Carbapenemase-producing Enterobacteriaceae (CPE) have spread globally since early 2010 in hospital facilities and have risen at an alarming rate in the human community [4,5].

CPE quickly disseminate resistant genes between bacteria through horizontal transfer, specifically plasmid-mediated gene transfer [6]. A plasmid is a mobile circular DNA carrying useful genes for adaptation and moving within and between species of bacteria. Inter-host transmission of resistant genes via plasmids enables the development of CPE cases in humans, not from using antibiotics directly, but from interacting with environments and hosts colonized with CPE [7]. As an illustration, plasmid-mediated, extended-spectrum β-lactamase-producing *Escherichia coli* (ESBL-EC) in the Dutch community is partly attributable to ESBL-EC in food, the environment, and animals [8].

AMR has rapidly disseminated worldwide in the community and hospitals due to excessive antibiotic usage, international travel, and global trade networks. The multiple sources of the AMR pandemic have prompted the European Union (EU), since 2010, to extend its surveillance of AMR to include food-producing animals. Cecal samples from live fattening pigs, veal calves, and broilers are collected at slaughterhouses and tested for resistant genes. Since 2016, this surveillance also includes CPE [9,10]. The current compulsory and harmonized AMR surveillance carried out by all EU member states is adequate to detect widespread AMR but will not quickly detect a newly emerging resistant bacterium due to the limited sample sizes and sampling frequency. In the current EU surveillance protocol, EU member states must annually collect a total of 170–300 samples, depending on the states’ production volume, from each species of food-producing animal. This sample size was set to detect CPE with 95% confidence, provided the prevalence is at least 2%. However, because the sampling is conducted only once a year, CPE could be widespread before they are detected. Enhancing EU surveillance to detect emerging CPE is possible through an increased sampling frequency, increased sample sizes, and risk-based surveillance.

This study aimed to inform risk-based surveillance for CPE *E. coli* (referred to as CPE in the remainder of the text of this paper) by ranking the farm types according to the likelihood of CPE introduction using a quantitative risk assessment model. We based our study on The Netherlands, but it is scalable to the European Union. We included six farm types at risk of CPE introduction: broiler farm, broiler breeder farm, fattening pig farm, breeding pig farm, farrow-to-finish pig farm, and veal calf farm. The reason for this selection was that these farm types are the ones most associated with AMR in The Netherlands [11]. Seven potential sources of CPE relevant to the Dutch livestock sector were identified in the literature review [7,12,13] Appendix A. These potential sources are hospital patients, returning travelers from abroad, companion animals, wild animals, wastewater from hospitals, imported livestock, and animal feed (Appendix A). The results from expert elicitation highlight returning travelers, wastewater from hospitals, and imported veal calves as the most important sources of CPE introduction (Appendix A).

## 2. Results

To estimate the risk of introduction, first, the number of farms exposed to CPE sources (Section 2.1) and the probability of colonization after exposure (Section 2.2) were estimated. These were combined into the risk of introduction by calculating the number of expected introductions (Section 2.3). The sensitivity of model output to model input parameters was determined by two methods of sensitivity analysis (Section 2.4). First, Spearman correlation coefficients were used to identify important uncertain parameters. Second, one-at-a-time sensitivity analysis was used to investigate the robustness of the ranking of risks to changes in each of the input parameters. Finally, different scenarios with respect to contamination of feed, restrictions on imports, and biosecurity were studied (Section 2.5).

### 2.1. Number of Farms Exposed to CPE

Based on our model calculations, fattening pig farms have the highest risk of CPE exposure, with over 600 farms in The Netherlands being exposed to at least one CPE source annually (Figure 1). The results indicate that 22% of the 2652 fattening pig farms and 12% of the 4513 pig farms (all farm types) in The Netherlands would be exposed to CPE. The numbers of broiler, breeding pig, and veal calf farms exposed to CPE is lower, though still considerable, with more than 100 farms exposed annually. The risk of CPE exposure is the lowest for broiler breeder farms with only 18 CPE expected exposures annually (Figure 1). The main sources of exposure are livestock feed, imported livestock, and returning travelers, while the small number of farms exposed to companion animals (four) and hospitalized patients is negligible (one).

### 2.2. Probability of Colonization Given Exposure to CPE

This probability was not calculated for imported livestock, since introduction of a colonized animal on the farm immediately results in colonization of the farm (where colonization of a farm was defined as the presence of at least one colonized animal on the farm). Livestock feed had the highest probability of colonization in the exposed farms (Table 1). Farm workers and veterinarians posed a very low probability of colonization to the exposed farms. The probability of colonization by exposure to companion animals was not calculated for the baseline scenario because we assumed that companion animals would not enter the barns, resulting in zero introduction to the small number of exposed farms. In the farm type comparison, exposed broiler and broiler breeder farms had the highest probability of colonization if exposed. The probability of colonization on a veal calf farm exposed to contaminated feed was the lowest of all farm types. The probabilities of colonization in veal calf and all three pig farm types exposed to CPE-colonized humans were equivalent. The probability of colonization was the lowest in all three pig farm types and veal calf fattening farms exposed to colonized returning veterinarians from overseas travel and hospital.

### 2.3. Ranking the Risk of Introduction: Combining Exposure and Colonization

The estimated number of fattening pig farms with CPE introduction was the highest, followed by broiler, fattening veal calf, and breeding pig farms (Figure 1). Farrow-to-finish farms and broiler breeder farms ranked lowest in terms of numbers of introductions. Exposure to contaminated feed was most likely to result in CPE introduction, with probabilities of colonization varying between 73% and 100% (Table 1). Exposure to hospitalized farm workers and returning travelers, on the contrary, was estimated to hardly ever result in CPE introduction to the farm due to a very low probability of colonization in exposed farms (Table 1). The expected annual number of CPE introductions to livestock farms in The Netherlands due to returning travelers was 5 × 10^−5^, which equals an introduction once every 20,000 years. For an individual farm, the estimated probability of colonization was highest on broiler farms (0.23, Table 2). Probabilities of colonization in fattening pig and farrow-to-finish farms were slightly lower (between 0.16 and 0.17). The probabilities of colonization in other farm types were lower than 0.1.

### 2.4. Result from Sensitivity Analysis

First, the Spearman rank correlation, a non-parametric metric between −1 and 1, was calculated for all input parameters with an uncertainty distribution to estimate the extent to which these input parameters determined the model results for each source (Section 2.4.1). Secondly, one-at-a-time (OAT) sensitivity analysis was performed (Section 2.4.2). In this additional sensitivity analysis, the value of a single input parameter was either increased or decreased. The outcome of each adjustment was compared to the baseline scenario to investigate the impact of all input parameters on the estimated number of introductions. OAT sensitivity analysis was performed separately for each source. Then, to evaluate if changes in input parameters would affect the ranking of sources, we compared the results of the OAT sensitivity analysis across sources (Section 2.4.3).

#### 2.4.1. Result from Spearman Rank Correlation

Based on the model results, feed is indicated as the main contributor of CPE introduction for all livestock farm types (Table 2). The Spearman rank correlation for this source revealed that the prevalence of CPE-colonized patients in Dutch hospitals ($P_{CPE_{NL}}),$ which was combined with *E. coli* prevalence to infer the prevalence of CPE in feed ($P_{CPE_{feed}})$, 50% infectious dose (*ID*50), and the average batch size of feed ($V_{batch})$ are inputs that are strongly correlated with the expected number of introductions from feed (Figure 2). However, these parameters are not expected to affect the ranking of farm types for their introduction risk because these inputs are identical for all farm types apart from 50% infectious dose (*ID*50), which differs between farm types (Appendix A). CPE prevalence in livestock i in country j ($P_{CPE_{A}})$ is highly correlated with the expected number of CPE introductions from imported animals to all farm types. Though CPE prevalence in humans ($P_{CPE_{NL}}\;and\;P_{CPE})$ is correlated with the number of introductions from both hospitalized patients and returning travelers, the average number of farmers per farm ($AVG_{farmers})$ and the probability of admission to hospital during travel $\left(P_{admit}\right)$ were more correlated with pig and veal calf farm introductions than CPE prevalence in the returning traveler source. Introductions from returning travelers and hospitalized patients were also correlated with input parameters for probability of colonization given exposure such as infectious dose at 50% colonization (*ID*50) and proportion of CPE transferred from fomite to finger and vice versa ($C_{tran_{E}}\;\mathrm{and}\;C_{tran_{A}}).$

#### 2.4.2. One-at-a-Time Sensitivity Analysis per Source

One-at-a-time sensitivity analysis of the input parameters for introduction by feed unveiled two parameters that had a huge impact on the estimated number of introductions in different farm types: the total number of animals in The Netherlands ($N_{animal})$ and the amount of feed consumed per animal per day ($C_{a})$ (Figure 3). The total number of farms ($N_{farm\;})$ was used twice in the model, i.e., to obtain the number of animals per farm and the number of farms exposed, which compiled into a lower effect toward introductions than the total number of animals in The Netherlands ($N_{animal})$ and the amount of feed consumed per animal per day ($C_{a})$. Parameters with the least impact on introduction in all farm types were the number of bacteria in contaminated feed ($Ecoli_{concF}$) and the median infectious dose (*ID*50). These two parameters were involved in calculating the probability of colonization in an exposed farm ($P_{col_{s}})$, while other parameters were involved in calculating the number of exposed farms ($N_{col_{s}})$.

Input values of three impactful parameters, namely, the total number of animals ($N_{animal}$), total number of local farms ($N_{farm})$, and grams of feed ingested per livestock per day ($C_{a})$, in the baseline model were compared across all farm types (Appendix A). Fattening pig farms had the highest total number of farms ($N_{farm})$ but a moderate total number of fattening pigs ($N_{animal}$) and grams of feed ingested per fattening pig per day ($C_{a})$ compared to other farm types. The high number of introductions to veal calf farms arose from imported livestock. Two essential parameters that directly facilitate introduction to fattening veal calf farms are CPE prevalence in the source country $\left(P_{CPE_{A}}\right)$ and the number of livestock i per shipment $\left(N_{size}\right)$ (Appendix A). When the number of livestock i per shipment was enhanced two-fold, the number of farms exposed was also enhanced two-fold (Appendix A). It should be noted that the number of livestock per shipment is directly correlated with the annual number of animals imported $\left(N_{imp}\right).$ However, a two-fold increase in the CPE prevalence in livestock in source countries $\left(P_{CPE_{A}}\right)$ increases the number of introductions only slightly because of the very low prevalence estimates based on the zero CPE cases in livestock (as reported by most source countries).

Fattening pig farms and veal calf farms remained the highest in farm types with introductions from livestock feed and imported livestock in the OAT sensitivity analysis. None of the OAT analysis resulted in increased introduction from human sources. However, one scenario of the OAT analysis indicated introduction to fattening pig farms from the companion animal source.

#### 2.4.3. One-at-a-Time Sensitivity Analysis between Sources

To evaluate if changes in input parameters would affect the ranking of sources, we performed a pairwise comparison of the results of the OAT sensitivity analysis of individual sources (Appendix A). For example, for the comparison of feed and imported livestock, we compared 15 outcomes (7 parameters that were both increased and decreased, and the baseline) of the feed source to 7 outcomes of the imported livestock source (3 parameters that were both increased and decreased, and the baseline). This resulted in a total of 105 combinations of outcomes including 1 combination of baseline parameters for both sources (Appendix A). Of all the other 104 outcome combinations, we recorded if the ranking of the sources was different from the comparison of the baseline parameters in both sources. Feed consistently ranked as the source with the highest expected number of CPE introductions in all farm types, except for veal calf farms, when comparing sensitivity tests across all sources (Appendix A). Forty-four percent of the adjusted input parameters resulted in a higher introduction from imported livestock to veal calf farms than feed. In the baseline model, the colonization risk of imported livestock and feed for veal calf farms was on the same order of magnitude, with the risk of feed being slightly higher, whereas for all other farm types, the risk of imported livestock was very low compared to feed (Figure 1). On the other hand, all sensitivity tests produced non-zero introduction from feed, while a small proportion of sensitivity tests (19%) resulted in negligible introduction from imported livestock to most farm types except fattening pig and veal calf farms. Imported livestock always had a higher introduction risk than returning travelers, hospitalized patients, and companion animals (Appendix A).

### 2.5. Result from What-If Analysis

The effects of higher contamination levels in feed, less strict biosecurity at the farm level, and a ban on livestock imports from countries sampling less than 100 animals for CPE surveillance were explored by adjusting input parameters and evaluating the model outcome (number of introductions) in what-if scenario analysis.

CPE was introduced into eight (one breeding, five fattening pig, and two veal calf) additional farms when the number of *E. coli* contaminations increased to the maximum limit for rejecting feed as given by GMP+. This addition is small compared to the 767 expected introductions in the baseline model (Table 3). Interestingly, banning imports from countries with a low surveillance level (less than 100 animals sampled) reduced the risk of introduction from imported livestock by 71%. Following a minor increase in introduction from companion animals in a flexible biosecurity scenario, companion animals would be reclassified from no risk to a low-risk source. Conversely, introduction from returning travelers and hospitalized patients remained negligible when the number of bacteria on a person’s palms increased four times due to non-compliance with hand hygiene protocols.

## 3. Discussion

This is the first risk assessment that quantifies the risk of CPE introduction into livestock farms in The Netherlands. The results indicate that fattening pig farms ranked the highest with respect to the expected annual number of CPE-colonized farms. However, when considering the probability of CPE introduction per individual farm, broiler farms have the highest introduction risk. Our model indicates that feed is a major potential source of CPE introduction, but this risk estimate has a high uncertainty. Imported livestock is indicated as an important CPE source specifically for veal calf farms. Other sources (companion animals, hospital patients, and returning travelers) were assessed to be of minor or negligible importance.

The number of exposed farms was most important in determining the introduction risk expressed as the expected number of colonized farms for high-rank sources (feed and imported livestock), due to the high probability of colonization upon exposure ($P_{col_{s}})$ in both sources (probability varying between 0.73 and 1 for feed (Table 1), probability of 1 for livestock imports). The probability of an individual farm exposed to CPE due to feed was similar in broiler, fattening pig, and farrow-to-finish farms (Table 2). This probability equaled the probability of receiving at least one CPE-contaminated batch of feed ($P_{CPE_{batch}}$). Although broilers require much less feed per animal than pigs due to their relatively small size, the number of broilers kept per farm is higher, resulting in a similar amount of feed delivered to all farm types.

The overall probability of introduction for an individual farm resulting from all sources was the highest in the broiler sector. If exposed to CPE, broilers have a higher probability of colonization than pigs and veal calves due to the very low median infectious dose (ID50) in broilers. This parameter mainly affected the colonization probabilities of farms exposed to CPE-colonized humans because, for this source, the dose to which the animals are exposed is low. With high exposure doses, as was the case with feed, the probabilities of colonization are high, even when the *ID*50 is high. The total number of CPE introductions is thus mainly determined by the total number of farms exposed to CPE given the high probability of colonization upon exposure by the two major sources (0.73–1 probability). Consequently, the effect of changing the probability of colonization is much smaller than that of changing the number of exposed farms.

According to our model, thirteen percent of Dutch farms are estimated to be colonized by CPE each year, mainly via feed, which is clearly an overestimation as such a percentage of farms being colonized would be detectable under the current national surveillance protocol [14,15]. Still, an undetected CPE presence in Dutch livestock is possible, as the current national surveillance protocol was designed to detect at least one colonized animal with 95% certainty, provided the prevalence is 1% [14]. However, this surveillance protocol does not take into account clustering of colonization at the farm level, which decreases the sensitivity of the surveillance. Furthermore, introductions could have escaped detection because most farms for meat production (broiler, fattening pig, and veal calf) apply an all-in-all-out system that produces more than one batch of livestock annually, while the national surveillance collects samples only once a year from a single animal per batch at slaughter from part of the farms. Thus, for each farm unit, multiple samples distributed over time are necessary to calculate an accurate prevalence [16].

In our calculation, a major source of CPE introduction is feed, although no carbapenemase-producing bacteria have been found thus far in feed. The probability that batches are CPE-contaminated and the concentration of CPE in contaminated batches were both inferred from the CPE prevalence among humans, *E. coli* prevalence in feed, and the ratios of CPE, ESBL, and other *E. coli* in water sources. Using these proxy measures introduces uncertainty in the calculations. Multiple studies, however, indicated the presence of *E. coli* in feed to be as prominent as Salmonella, which is a major hazard in animal feed [1,17,18,19,20,21]. Despite no CPE detection in livestock feed, a small percentage of *E. coli* from feed collected in Portugal and the United States carried resistant genes against ampicillin and cefotaxime [19,22,23]. It is, therefore, reasonable to assume that CPE contamination of feed is possible. Although halving the CPE prevalence in feed lowered the risk of feed considerably (Appendix A), feed still remained an important source of CPE introduction, still being higher than the risk of imported animals. It is therefore recommended to investigate this source of CPE in more detail to either discard this source as a risk or to enable mitigation strategies.

The probability of batches of feed contaminated with CPE ($P_{CPE}{}_{feed}),$ the number of batches delivered to a farm each year ($N_{batch}$), the median infectious dose (ID50), and the concentration of CPE *E. coli* (cfu/g) in contaminated animal feed ($CPE_{concF}$) are four parameters worth further examination because they had a large impact on the introduction risk and are surrounded by considerable uncertainty. Uncertainty in the probability of batches of feed contaminated with CPE ($P_{CPE}{}_{feed}),$ and the concentration of CPE *E. coli* (cfu/g) in contaminated animal feed were due to lack of data for CPE, and these parameters were therefore inferred from the prevalence and concentration of *E. coli* in feed and other sources. Equally, no data were available on the median infectious dose (ID50) for CPE in livestock, and therefore estimates from studies on ESBL in broilers and pigs were used. Uncertainty in the number of batches delivered to a farm each year ($N_{batch}$) stems from generalizing highly variable parameters into an average value. The impact of overestimating these parameters was assessed in a sensitivity analysis, where the number of introductions from feed was reduced by, at most, 47% (Appendix A). Still, the 47% reduction in the number of introductions from feed remains higher than other sources (Appendix A).

Whereas most farm types have a low risk of introduction via routes other than feed, veal calf farms have a high risk of introduction by imported animals. Farms received a higher number of batches of imported veal calves than other animal types due to a high number of imported animals and small batch sizes. Furthermore, the inferred CPE prevalence in veal calves in source countries ($P_{CPE_{A}})$ is higher than the estimated CPE prevalence in pigs and broilers [9,24]. Eighteen EU member states did not collect any samples from veal calves for CPE surveillance (Appendix A). Therefore, the CPE prevalence in veal calves in these member states was inferred from ESBL surveillance in bovine meat (Appendix A), resulting in a higher CPE prevalence in our calculations for veal calves. Both countries from which a high number of veal calves are imported ($N_{A})$ and countries with a high inferred probability that imported veal calf batches are colonized with CPE ($P_{CPE_{A}})$ (Appendix A) have a high risk of CPE introduction. This outcome resembles a risk assessment by EFSA, which concluded that EU member states with higher volumes of livestock trading have a higher risk of disseminating AMR-ESBL bacteria [2,25]. We believe that the high risk level expected for veal calves from the model could be an overestimation given the lack of CPE detection in veal calves in EU surveillance (EARS-net). The high prevalence estimates for source countries were thus not based on reported detections but resulted from uncertainty due to low sample sizes. However, CPE cases in cows were detected in European countries [26], and imported veal calves were ranked first for risk of CPE in our expert elicitation (Appendix A). The scenario of reducing risk by only allowing countries that sample more than 100 animals annually to export to The Netherlands was shown to be an effective mitigation strategy in the what-if analysis. The expected number of introductions was reduced by 71%. It should, however, be kept in mind that this strategy reduces the potential CPE introductions resulting from uncertainty in CPE prevalence in veal calves in source countries. Countries with an effective surveillance program in calves that do find CPE in calves might, in reality, pose a higher risk to the Dutch veal calf sector. A more reliable estimate of the CPE introduction risk via imported livestock can be obtained via enacting EU-wide mandatory surveillance with enough samples in all countries exporting veal calves to EU member states.

Humans were initially thought to be a high-risk source because of high numbers of overseas travel and CPE presence in hospitals [4], but the risk of these sources was found to be very low. In spite of a non-zero number of farms exposed to returning travelers and hospitalized patients (the probability of exposure of an individual farm is as high as for imported livestock (Table 2)), the extremely small calculated dose of CPE ingested by livestock leads to a very low number of expected colonizations in the exposed farms (Table 1). The prevalence of the clinically relevant CPE Klebsiella pneumoniae in humans is slightly higher than CPE *E. coli* [10]. Only the latter was considered in this risk assessment. Including CPE Klebsiella pneumoniae is, however, not expected to result in a change in the ranking of sources given the huge difference in the estimated risk between feed and imported livestock, on the one hand, and travelers and hospitalized patients, on the other. Likewise, CPE introduction from the companion animal source was assessed to be negligible because there is no exposure of farm animals to colonized companion animals if strict biosecurity is applied. What-if analysis evaluated the effect of reduced biosecurity in farms, where hand hygiene and exclusion of companion animals from the barns were not complied with [27,28,29,30,31]. This scenario still resulted in a very low number of expected introductions from human and companion animal sources. This is explained by the low number of humans and companion animals attributed per farm and the very low probability of colonization of the farm if exposed to CPE-colonized humans or companion animals.

The outcome of this introduction risk assessment was used to rank farm types and sources of their CPE introduction risk. The results for the absolute numbers of exposures and introductions have a large uncertainty and cannot be viewed as accurate quantitative risk estimates. The results of the sensitivity analysis provide good indications of the uncertain input parameters that have the largest impact on the model results. Parameters with both a large uncertainty and a large impact are important knowledge gaps that can be targeted in future studies. Despite these uncertainties, the ranking of farm types and sources was robust and the outcome of this risk assessment can thus be used for targeted CPE surveillance [32,33,34].

## 4. Materials and Methods

We quantitatively assessed the risk of CPE introduction to broiler, pig, and veal calf farms from five potential CPE sources, i.e., imported livestock, livestock feed, companion animals, hospital patients, and returning travelers, and ranked farm types by the expected number of farms with CPE introduction and the probability of CPE introduction for an individual farm. This quantitative risk assessment followed the guidelines for import risk assessment provided by the World Organisation for Animal Health (OIE) [32,33] to assess the risk of exposure of farms, and the guidelines for microbial risk assessment provided by the Codex Alimentarius to assess the risk of infection upon exposure [35,36]. We conducted sensitivity analysis to assess the effect of uncertainty surrounding important input parameters toward the output and evaluated alternative biosecurity practices and trade restrictions via scenarios analysis.

Despite being highlighted as an important potential CPE source, wastewater from hospitals was excluded from the model because CPE will be effectively removed in the wastewater treatment facilities. Additionally, although small traces of CPE could be present in surface water due to overflow from rainfall, the vast majority of the meat-producing animals of our concern (veal calf, fattening pig, breeding pig, broiler, and broiler breeder) were raised in a closed system where they drink tap water. This water source undergoes extensive purification, ensuring no traces of resistant bacteria such as CPE [37,38,39]. Wild mammals and birds were also excluded from the model. Small mammals such as rodents move locally and thus would not be exposed to CPE from outside The Netherlands. Interactions between local target farms and wild birds are mostly prevented as livestock live in closed barns.

### 4.1. Risk Model

#### 4.1.1. Model Outline

CPE introduction was defined as the colonization of at least one animal with CPE upon exposure of a farm to any of the sources included in the model. The risk of CPE introduction was modeled with two submodels (Figure 4). The first submodel used scenario tree modeling to estimate the number of farms exposed to CPE-colonized sources ($N_{col}$). The second submodel was a microbial risk assessment model to estimate the probability that at least one animal will be colonized on an exposed farm ($P_{col}$) given the dose to which the animals on the farm are exposed ($CPE_{ing}$), using an exponential dose–response model. The outputs of both submodels were combined to calculate the expected annual numbers of farms on which CPE is introduced ($N_{intro}$). Parameters and values used in the model are presented in Table 1.

The annual expected number of CPE introductions via each source was calculated using multiple input parameters, some of which are uncertain. Parameters on CPE prevalence, CPE concentration, number of animals in transport, and colonization duration were chosen to be included with a distribution to account for uncertainty and variability. Less variable data, such as total numbers of farms and livestock in The Netherlands, were entered as point estimates. The impact of these parameters on the model results was studied by a sensitivity analysis where the input values were increased and decreased two-fold. We ran 10,000 iterations using Monte Carlo sampling in ModelRisk, an add-on for Microsoft Excel version 1908^®^ [40].

#### 4.1.2. Submodel I: Scenario Tree Model

The exposure of the following six farm types: broilers, broiler breeders, fattening pigs, breeding pigs, farrow-to-finish, and veal calves, to CPE from sources s (imported livestock (A), livestock feed (F), companion animals (C), farm workers being hospitalized (H), and farm workers traveling abroad (T)) was calculated by multiplying the number of farms in contact with people or animals or receiving feed, $N_{s},$ or by the probability that these persons or animals are colonized with CPE, or that the feed is contaminated with CPE, $P_{CPE_{s}}$. Mixed species livestock farms were not considered in the risk assessment because they represented a small proportion of local farms [41].
(1)$$N_{col_{s}}=N_{s}\cdot P_{CPE_{s}}$$

##### Imported Livestock

The number of farms exposed to CPE from imported animals, $N_{col_{A}}$, was calculated by multiplying the annual number of batches of animals imported from the source country—among all EU member states in 2017—to six farm types ($N_{A})$ by the probability that an imported batch from the source country which is delivered to an individual farm type is colonized with CPE ($P_{CPE_{A}})$.

We assumed that CPE colonization is maintained during transport and will reach local farms without detection. Sustained CPE colonization in animals during transportation between EU member states is likely within the maximum 24 h transport time [42], because in livestock, ESBL colonization can be maintained for 30 to 180 days [43,44,45,46]. Within the EU, antimicrobial testing in imported animals is not obligatory and not conducted [2]. The probability of detecting a CPE-colonized animal is thus negligible and was not accounted for in the calculations.

##### Livestock Feed

The number of farms exposed to CPE-colonized feed,$\;N_{col_{F}}$, was calculated as the product of the total number of six farm types in The Netherlands ($N_{farm}$) and the probability that an individual farm would receive at least one batch of feed contaminated with CPE $\left(P_{CPE}{}_{batch}\right)$. $P_{CPE}{}_{batch}$ was calculated from the probability that a batch of feed is contaminated with CPE ($P_{CPE_{feed}})$ and the annual number of feed batches received by a farm ($N_{batch})$. The estimated value for $P_{CPE_{feed}}$ was used for all farm types because no data were available to estimate $P_{CPE_{feed}}$ separately for each farm type.
(2)$$P_{CPE_{batch}}=1-{\left(1-P_{CPE_{feed}}\right)}^{N_{batch}}$$

##### Companion Animals

The number of farms exposed to CPE-colonized companion animals ($N_{col_{C}})$ was derived by multiplying the number of farms with companion animals$\;\left(N_{C}\right)$ by the probability that companion animals in The Netherlands are colonized with CPE $\left(P_{cCPE_{NL}}\right)$. The number of farms having companion animals$\;(N_{C}$) was calculated from the total number of farms ($N_{farm})$ multiplied by the probability of farms having a companion animal$\;(P_{farmC}$).

##### Farm Workers

CPE introduction from humans is possible when farm-related workers k (farmers, veterinarians) acquire CPE during holidays outside The Netherlands or in local hospitals (Figure 5). Here, the number of farm workers acquiring CPE in hospital $(N_{colH_{k}}$) was calculated by multiplying the number of farm workers hospitalized ($N_{H})$ by the probability that patients acquire CPE in Dutch hospitals$\;\left(P_{CPE}{}_{NL}\right)$. The number of farm workers hospitalized$\;\left(N_{H}\right)$ was estimated by multiplying the number of farm workers and veterinarians in The Netherlands$\;(N_{k}$) by the annual probability of hospital admission in the general population ($P_{admit_{NL}}$).

The number of farms exposed to CPE through infected farm workers returning from travel abroad$\;(N_{colT_{k}}$) was calculated by multiplying the number of farm workers returning from abroad$\;(N_{T_{k}}$) by the probability of travelers acquiring CPE during travel. The probability of traveler-acquired CPE differed according to the 16 regions of destination based on the United Nations geoscheme excluding The Netherlands [47] (Appendix A), and therefore calculations were performed for each region individually. The number of farmers returning from each of these regions was estimated based on the probability of Dutch travelers visiting each region $\left(P_{T}\right)$. Both the probability of acquiring CPE in the hospital $\left(P_{CPE}\right)$ and the probability of acquiring CPE from the community $\left(P_{cCPE}\right)$ during travel were considered in the model. The probability of hospital-acquired CPE during holidays $\left(P_{CPE}\right)$ was multiplied by the probability of travelers being hospitalized $\left(P_{admit}\right)$. The probability of community-acquired CPE $\left(P_{cCPE}\right)$ was multiplied by the probability of non-hospitalized travelers ($1-P_{admit})$ (Figure 2). The estimated value for was used for all regions because no data were available to estimate $P_{admit}$ separately for each region.

#### 4.1.3. Submodel II: Exposure Assessment

We estimated the numbers of farms where CPE was introduced by multiplying the number of exposed farms ($N_{col_{s}}$) by the probability that at least one animal on an exposed farm would become colonized ($P_{col_{s}})$. The probability that at least one animal on an exposed farm would become colonized was calculated with an exponential dose–response model using the total number of CPE *E. coli* bacteria ingested by the animals on the farm ($CPE_{ing}{}_{s})$ as the dose. The ingested dose ($CPE_{ing}{}_{s})$ was calculated separately for each farm type and CPE source s, as described in Equations (3)–(5). These calculations were not performed for the source imported livestock, since the introduction of a colonized animal into a livestock farm directly results in a colonized farm.

##### Animal Feed

The ingested dose of CPE from contaminated feed on a single farm $\left(CPE_{ing_{F}}\right)$ was estimated as the product of the concentration of CPE *E. coli* (cfu/g) in contaminated animal feed delivered to a farm $\left(CPE_{concF}\right)$ and the average weight of one batch of feed in grams $\left(V_{batch}\right)$.
(3)$$CPE_{ing_{F}}=CPE_{concF}\cdot \;V_{batch}\;$$

##### Companion Animals

To estimate the total CPE deposited by companion animals in the farm environment, we multiplied the concentration of CPE in companion animal feces ($CPE_{gramC})$ (cfu/g) by the average weight (grams) of feces defecated by a companion animal in each defecation ($W_{fec})$, the daily defecation frequency of companion animals ($N_{eli_{C}}$), the length of the colonization period in companion animals in days ($T_{CPE_{C}}$), and the proportion of time that a companion animal is present in the barn ($P_{barnC}{}_{i})$. The total CPE ingested by the farm animals ($CPE_{ing_{C}}$) was subsequently calculated by multiplying the deposited CPE in the farm environment by the proportion of excreted bacteria taken up by the livestock animals from the farm environment ($C_{tranA})\;$(Table 1).
(4)$$CPE_{ing_{C}}=W_{fec}\cdot N_{eli_{C}}\cdot T_{CPE_{C}}\cdot CPE_{gramC}\;\cdot \;P_{barnC}\cdot C_{tranA}\;$$

##### Farm Workers

The number of CPE bacteria ingested by colonized farm workers$\;\left(CPE_{ing_{H}}\right)$ was calculated in a similar manner to the ingested dose from companion animals$\;\left(CPE_{ing_{C}}\right)$, albeit with different inputs. The transmission event started after the colonized farm worker (farmer or veterinarian) used the toilet for defecation. We assumed CPE contaminated their hands after toilet usage and that not all would be removed by hand washing. Thus,$\;CPE_{hand}$ was the number of CPE (cfu) remaining on a farm worker’s hands after hand washing. The number of CPE deposited in the farm environment was then calculated by multiplying this number by the daily defecating frequency of humans ($N_{eli_{H}}$), the length of the colonization period of CPE in humans in days ($T_{CPE_{H}}$), the proportion of bacteria transferred from the farm worker’s hand to the farm environment ($C_{tran_{E}}$), and the proportion of the day that a worker is in the barn ($P_{barnH}$). The last parameter is different between farm workers and veterinarians, assuming that a farmer spends much more time in the barn of a single farm than a vet. The total CPE ingested by the farm animals ($CPE_{ing_{H}}$) was subsequently calculated by multiplying the deposited CPE in the farm environment by the proportion of bacteria taken up by the livestock animals from the farm environment ($C_{tranA})$.
(5)$$CPE_{ing_{H}}=CPE_{hand}\cdot N_{eli_{H}}\cdot T_{CPE_{H}}\cdot C_{tran_{E}}\cdot \;P_{barnH}\cdot \;C_{tran_{A}}$$

#### 4.1.4. Submodel II: Dose–Response Model

The probability that at least one animal at farm type i is colonized with CPE ($P_{col_{s}})$ is a function of the CPE ingested dose from a source s ($CPE_{ing}{}_{s}$) and the dose–response parameter. The dose–response parameter gives the probability of a single CPE bacterium colonizing an animal’s gut (P) and is calculated from the *ID*50 (the dose at which 50% of the animals are expected to be colonized). An exponential dose–response model was used, and P was calculated as $\frac{ln2}{ID50}$. The probability that at least one animal is colonized with CPE was then calculated as
(6)$$P_{col_{s}}=1-e^{-\left(P\;\cdot \;CPE_{ing_{s}}\right)}$$

#### 4.1.5. Risk Estimate Combining Submodel I and Submodel II

The expected number of introductions to each farm type from each source s$\;(N_{intro_{s}}$) was calculated by multiplying the number of farms exposed to each source s$\;\left(N_{col_{s}}\right)$ by the probability that at least one animal on an exposed farm is colonized$\;\left(P_{col_{s}}\right)$.
(7)$$N_{intro_{s}}=N_{col_{s}}\cdot P_{col_{s}}$$

The absolute risk of CPE introduction into local Dutch farms was given as the expected annual number of introductions per farm type $(N_{intro}$) from all CPE sources considered in the model. The probability of CPE introduction for an individual farm was estimated by dividing the number of expected introductions per farm type by the total number of farms of this type in The Netherlands.

### 4.2. Input Parameters

#### 4.2.1. Imported Livestock

Data on the number of livestock imported into The Netherlands from EU member states ($N_{imp})$ were available for the period 2016 to 2020 and fluctuated slightly. Import data for the year 2017 were used in the baseline model to be consistent with the data used for the number of farms and veterinarians. The livestock import records were derived from two publicly available sources, namely, Statistics Netherlands (CBS) and The Netherlands Enterprise Agency (RVO) (Appendix A and Table 4) [48]. To estimate the number of imported batches $\left(N_{A}\right)$, the annual number of imported animals was divided by the average number of livestock per shipment $\left(N_{size}\right)$. In estimating the number of animal batches delivered to each farm type annually $\left(N_{batch}\right)$, we assumed that all imported one-day-old broilers would go to broiler farms, all imported parent broilers would go to broiler breeder farms, all imported veal calves would go to veal calf farms, all imported piglets would go to fattening pig farms, and all imported breeding pigs would go to breeding pig farms and farrow-to-finish pig farms in a ratio of 2:1, representing the ratio of these farms in The Netherlands.

The probability that imported animals from EU member states are colonized with CPE ($P_{CPE_{A}}$) was directly inferred from national surveillance data provided by the European Antimicrobial Resistance Surveillance Network [9,24]. CPE surveillance in livestock consisted of random sampling of fecal samples from live animals at slaughter, the results of which were used as a proxy for herd prevalence in the risk model. Data on surveillance in pigs and broilers were available for all EU member states, EFTA countries, and the UK, whereas only 9 EU member states and 2 EFTA countries (Norway and Switzerland) reported on CPE surveillance in calves. For countries that had no data on surveillance in calves, the probability of CPE colonization was inferred from the surveillance in bovine meat (Appendix A). The probability that imported animals are colonized with CPE ($P_{CPE_{A}}$) was estimated using a beta distribution based on the number of animals sampled (n), the number of animals that tested positive (s), and test sensitivity (se) (Table 4).

#### 4.2.2. Animal Feed

The average number of batches of feed received by individual farms ($N_{batch}$) was calculated as
(8)$$N_{batch}=\frac{n_{a}\cdot c_{a}\cdot 365}{V_{batch}}$$
where $n_{a}$ is the average number of animals on a farm of type *i*, $c_{a}$ is the average consumption of feed per day per animal on each farm type (in grams), and $V_{batch}$ is the average size of a batch of feed delivered to a farm (in grams). The average number of animals on farm type *i* ($n_{a})$ was calculated by dividing the total number of animals in The Netherlands present at each farm type ($N_{animal}$) by the total number of farms at each farm type in The Netherlands ($N_{farm}$). The number of Dutch farms ($N_{farm}$) and livestock heads ($N_{animal}$) was based on 2017 data provided by Statistics Netherlands. Due to a lack of farm-specific data, $V_{batch}$ was set equal for all farm types.

Since feed ingredients are heat-treated, CPE contamination was expected to result from cross-contamination during processing and storage in a local feed mill. The probability of feed colonized with CPE was therefore based on Dutch data. As there is no CPE surveillance conducted on animal feed at all, the probability of batches of feed contaminated with CPE ($P_{CPE}{}_{feed})$ was inferred from the ratio between *E. coli* prevalence in feed ($P_{ec_{feed}}$) and in humans ($P_{ec_{NL}}$) under the presumption that the ratio of *E. coli* in the two aforementioned sources is the same as the CPE ratio (Equation (9)). $P_{ec_{feed}}$ was based on the prevalence of compound feed for cattle contaminated with *E. coli* in the EU [23], and $P_{ec_{NL}}$ was based on the prevalence of *E. coli* in Dutch residents reported in the national surveillance of antimicrobial resistance [11]. No data were available for the CPE prevalence in the Dutch community ($P_{cCPE_{NL}}$). However, we had data on CPE prevalence in Dutch hospitals ($P_{CPE_{NL}}$). Therefore, $P_{cCPE_{NL}}$ was inferred from the ratio between ESBL *E. coli* in the community and in clinical settings $\left(C_{com:\;cli}\right)$, under the presumption that the CPE correlation between the community and the clinical setting is similar to the ESBL *E. coli* correlation in European countries. The CPE prevalence in Dutch hospitals ($P_{CPE_{NL}})$ was therefore multiplied by the ratio of ESBL *E. coli* in the community versus ESBL in a clinical setting,$\;C_{com:\;cli}.$ This ratio was estimated to be 0.79 based on the Pearson correlation between ESBL prevalence in the community and in the clinical setting in the EU, as observed in five studies [49,50,51,52,53]. The derived value of $P_{CPE_{feed}}$ was used for all farm types owing to the lack of data on *E. coli* in feed for other animal species.
(9)$$P_{CPE_{feed}}=\frac{P_{cCPE_{NL}}}{P_{ec_{NL}}}\cdot P_{ec_{feed}}$$

No data were available on the concentration of CPE in feed if it was contaminated. The concentration of CPE in feed ($CPE_{concF}$) was estimated by multiplying the strict concentrations of *E. coli* allowed (minimum rejection limit) in feed components $\left(Ecoli_{concF}\right)$ as given by GMP+ [54] by the ratio of *E. coli* carrying CPE genes to non-resistant *E. coli* ($P_{CPE:EC}$), as observed in samples from 100 Dutch wastewater treatment facilities [37].

#### 4.2.3. Companion Animals

The number of farms with a companion animal $\left(N_{C}\right)$ was calculated by multiplying the total number of farms in The Netherlands $(N_{farm}$) by the proportion of farms with companion animals $\left(P_{farmC}\right)$. No data were available on the proportion of farms with companion animals in The Netherlands. Assuming that farmers’ behavior in The Netherlands does not greatly deviate from other Western regions, we used surveillance data of farmers’ behavior in the United States of America to estimate $P_{farmC}$.

The probability of companion animals colonized with CPE in The Netherlands was set equal to the CPE prevalence in the Dutch community ($P_{cCPE_{NL}})$. Although some information on numbers of colonized companion animals in The Netherlands was available from the Monitoring of Antimicrobial Resistance and Antibiotic Usage in Animals in The Netherlands report [55], these numbers were not considered representative as these were cases from animals visiting a veterinary clinic only (Appendix A). The concentration of CPE (cfu/g) in feces ($CPE_{gramC})$ was estimated from the concentration of ESBL *E. coli* (cfu/g) in animal feces ($ESBL_{gramFec})$ measured in an observational study of healthy dogs in the United States [56] and the proportion of ESBL *E. coli* carrying CPE genes ($P_{CPE:ESBL}$) [37].

The frequency of defecating ($N_{eli}{}_{C})$ was based on a report from a commercial feed company in the United Kingdom [57]. The weight (grams) of feces defecated by a companion animal was based on a study in healthy medium-sized dogs in the United States ($W_{fec})$ [58]. Time spent in the livestock area $(P_{barnC}$) was set to zero for all farm types in the default calculations, assuming compliance with biosecurity protocols in The Netherlands. However, we explored non-zero $P_{barnC}$ reflecting farms with a lower biosecurity standard in a what-if analysis (Section 2.5 & Table 3). The proportions of CPE transfer from the environment to animal ($C_{tranA}$) were based on a study that measured the proportion of *Acinobacter* transferred from fomite to finger [59]. The CPE colonization period in companion animals ($T_{CPE_{C}})$ was set equal to the ESBL *E. coli* colonization period in healthy dogs in The Netherlands [60].

#### 4.2.4. Farm Workers

The total number of farms in The Netherlands $\left(N_{farm}\right)$ was multiplied by the average number of employees per farm ($Avg_{farmer}$) to parameterize the number of farmers ($N_{farmers}).$ Each farm is typically visited by a single veterinarian, and therefore the number of veterinarians ($N_{vet})$ in the model was set equal to the total number of farms in The Netherlands $\left(N_{farm}\right)$. The number of farm-related workers spending their holiday abroad $(N_{T_{k}}$) was calculated by multiplying the number of farm workers ($N_{farmer})$ and veterinarians ($N_{vet}$) by the probability of farm workers and veterinarians traveling abroad for their holidays ($P_{holiday})$. The probability of farmers taking a holiday abroad was derived from an online survey among 300 Dutch farmers conducted by a farm-oriented magazine, Boerderij (Farm) [61]. The probability of veterinarians taking a holiday abroad was based on data from Statistics Netherlands [41] for the general Dutch population. The proportion of Dutch travelers visiting each UN region $\left(P_{T}\right)$ was based on Statistics Netherlands data from 2013, where the number of holidays to each region was divided by the total number of holidays taken by Dutch citizens (Appendix A). To estimate the probability of hospital admission for farm workers ($P_{admit_{NL}}$), the number of Dutch inpatients in 2017 was divided by the total population of The Netherlands in 2017. The prevalence of CPE in hospital $\left(P_{CPE_{NL}}\right)$ was based on data provided by EARS-Net [10]. The probability of hospital admission during holidays outside of The Netherlands ($P_{admit}$) was derived from a study among 2000 Dutch travelers. The probability of acquiring CPE during hospitalization $\;\left(P_{CPE}\right)$ in non-European countries was parameterized from national surveillance on CPE prevalence from multiple countries around the world reported in the WHO’s global report of surveillance [62] and independent academic publications [63,64]. The probability of non-hospitalized travelers acquiring CPE from the community in a foreign country $\left(P_{cCPE}\right)$ was inferred by multiplying the hospital CPE prevalence ($P_{CPE})$ by the ratio of ESBL in the community versus ESBL in the clinical setting $\left(C_{com:\;cli}\right)$ (Appendix A). The number of CPE (cfu) remaining on a farm worker’s hands after hand washing ($CPE_{hand}$) was estimated from an observational study in Mexico among tomato farmers, in which the number of *E. coli* on hands after toilet use followed by hand washing ($Ecoli_{hand})$ was measured. $Ecoli_{hand}$ was multiplied by the probability of *E. coli* carrying CPE genes ($P_{CPE:EC}$) to calculate CPE (cfu) on farm workers’ hands. The number of defecations per day ($N_{eli_{H}}$) was retrieved from an observational study of 2000 returning Dutch travelers (Arcilla et al., 2016). Proportion of time spent in the livestock area $\left(P_{barnH}\right)$ was estimated at eight hours a day for farmers and one hour per week for veterinarians. The proportions of CPE transfer from the hands to the environment ($C_{tranE})$ were based on the same study used to estimate the proportions of CPE transfer from the environment to the animal ($C_{tranA}$) [59].

#### 4.2.5. Dose–Response Parameter

The median infectious dose (ID50) was used to calculate the dose–response parameter (P). The median infectious dose (ID50) was based on experimental studies for ESBL in broilers and pigs. No data were available to estimate the ID50 for veal calves, and, therefore, it was set equal to the median infectious dose of pigs.

**Table 4 antibiotics-11-00281-t004:** Input parameters for the model to assess the risk of CPE introduction into Dutch livestock farms.

Input *	Description	Value Distribution **	Value in Sensitivity Analysis	References
$N_{intro}$	Expected annual number of farms on which *CPE* is introduced			
$N_{col_{s}}$	Number of farms exposed to *CPE*-colonized sources *s* (imported livestock (A), livestock feed (F), companion animals (C), farm workers being hospitalized (H), and farm workers traveling abroad (T))			
$N_{s}$	Number of farms in contact with people, import animals, companion animals, and livestock feed			
$P_{CPE_{s}}$	Probability of sources exposed to farm are colonized/contaminated with *CPE*			
$P_{CPE}{}_{batch}$	Probability that an individual farm receives at least one batch of feed contaminated with *CPE*			
$N_{batch}$	Annual number of feed batches received by a farm			
$P_{CPE_{feed}}$	Probability that a batch of feed is contaminated with *CPE*			
$N_{C}$	Number of farms with companion animals			
$N_{H}$	Number of farm workers/vets hospitalized			
$N_{T_{k}}$	Number of farm workers/vets returning from abroad			
$CPE_{ing}{}_{s}$	Total number of CPE *E. coli* bacteria ingested by the animals on an exposed farm			
$CPE_{concF}$	Total number of *CPE E. coli* (cfu/g) in contaminated animal feed			
$CPE_{gramC}$	Total number of *CPE E. coli* (cfu/g) in companion animal feces			
$CPE_{hand}$	Total number of *CPE E. coli* (cfu) remaining on a farm worker’s hands after hand washing			
P	Probability of a single CPE bacterium colonizing an animal’s gut			
$N_{imp}$	Annual number of imported broilers, parent broilers, piglets, breeding pigs, and veal calves from EU member states *j* to farm type *i* in The Netherlands	Appendix A	Yes	[41,48]
se	CPE surveillance sensitivity	0.85	Yes	[14]
$P_{CPE_{A}}$ $P_{CPE_{NL}}$	*CPE* prevalence in livestock *i* in country *j* *CPE* prevalence in hospitalized patients in The Netherlands	Beta (α/se, β) (values of beta distribution in EFSA reference) Beta (8/se, 6676)	Yes	[9,10,24]
$P_{CPE}$	CPE prevalence in hospital patients in region *m*	Beta (α/se, β) (values of beta distribution are in Appendix A)	Yes	[63,64,65,66,67,68,69,70,71,72,73,74,75,76,77,78,79,80,81,82,83]
$C_{com:\;cli}$	Ratio of ESBL in the community versus ESBL in a clinical setting	0.79	N	Appendix A
$P_{ec_{feed}}$	Prevalence of *E. coli*-contaminated feed in compound cattle feed	Beta (59, 46)	Yes	[23]
$P_{ec_{NL}}$	Prevalence of *E. coli* in Dutch residents	Beta (159,620, 280,677)	Yes	[55]
$N_{size}$: broiler $N_{size}$: piglet $N_{size}$: breeding pig $N_{size}$: veal calf	Number of livestock *i* per shipment	Pert (45,00,47,000, 55,000) Pert (100, 260,300) Pert (65, 80, 95) Pert (30, 150, 200)	Yes	[29]
$N_{farm}\mathrm{and}$ $N_{animal}$	Total number of farm types *i* and total number of animals *i* in The Netherlands	Appendix A	Yes	[41]
$N_{K}$	Total number of farm workers and veterinarians in The Netherlands	Appendix A	Yes	[41]
$c_{a}$	The average grams of feed consumed by livestock *i* per day	Appendix A	Yes	[84,85,86]
$V_{batch}$	The average grams of feed delivered to a farm derived from the volume of a standard transport truck	Pert (3 × 10^6^, 16 × 10^6^, 3 × 10^7^)	Yes	[29]
$Ecoli_{concF}:$ broiler $Ecoli_{concF}$: fattening pig $Ecoli_{concF}$: breeding pig $Ecoli_{concF}$: veal calf	Concentrations of *E. coli* in feed components following minimum rejection limit by GMP+ (cfu/g)	11.8 11.8 14.3 7.3	Yes	[54]
$Ecoli_{hand}$	The amount of *E. coli* remaining on a farm worker’s hands after toilet use and subsequent hand washing (cfu)	Log-normal (63, 5.02)	Yes	[28]
$ESBL_{gramFec}\;$(cfu/g)	Number of *E. coli* (cfu) in a gram of healthy companion animal’s feces	Normal (70, 35)	Yes	[87]
$P_{CPE:EC}$ $P_{CPE:ESBL}$	Proportion of *E. coli* carrying CPE genes and proportion of ESBL *E. coli* carrying CPE genes	0.00004 0.00424	N	[37]
ID50: broiler ID50: pig and veal calf	Infectious dose of ESBL *E. coli* at which, on average, 50% of livestock species *i* are colonized (cfu)	Log-normal (5, 5) Log-normal (4695, 9187)	Yes	[56,88,89]
$P_{farm_{C}}$	Proportion of farms that have companion animals	Beta (298, 148)	Yes	[56]
$W_{fec}$ (grams)	Grams of feces defecated by a companion animal in one defecation	Normal (70, 35)	Yes	[58]
$N_{eli_{C}}$ $N_{eli_{H}}$	The average number of defecations by companion animals and humans per day	Pert (1, 2, 5) Uniform (1,3)	Yes	[57] Assumption
$T_{CPE_{C}}$ $T_{CPE_{H}}$	Colonization duration of CPE in companion animals and humans (days)	Pert (0, 120, 180) Pert (1, 30, 365)	Yes	[60,90]
$P_{barnC}$ $P_{barnH}$: farm worker $P_{barnH}$: veterinarian	Proportion of day a companion animal, farm worker, and veterinarian spent in the barns	0 0.33 0.005	Yes	Assumption
$C_{tranA}$ $C_{tranE}$	Proportion of *Acinobacter* transferred from fomite to finger (A) and from finger to fomite (E)	Log-normal (0.24, 0.14) Log-normal (0.06, 0.06)	Yes	[59]
$P_{T}$	The probability of Dutch travelers visiting 16 world regions in 2013	Appendix A	Yes	[41]
$P_{holiday}$: broiler and pig farm worker $P_{holiday}$: veal calf farm worker $P_{holiday}$: veterinarian	Probability of farm worker on farm *i* taking holiday abroad annually	0.53 0.33 0.64	Yes	[41,61,91]
$Avg_{farmers}$	The average number of farm workers in all farm types	Pert (1, 2, 4)	Yes	Assumption
$P_{admit}$ $P_{admit}{}_{NL}$	Probability of hospital admission while traveling overseas and in The Netherlands	0.04 0.054	Yes	[41,90,92]

Footnotes: * Type of farm is indicated by subscript *i* and source country by *j*. **** Parameters for input distributions given in brackets: beta (α,β), where α equals the number of positives plus one, and β the number of negatives plus one; log-normal (mean, SD); normal (mean, SD); pert (minimum, most likely, maximum); uniform (minimum, maximum). Parameters with an empty *Value Distribution* are parameters calculated from the raw input.

### 4.3. Sensitivity Analysis

#### 4.3.1. Spearman Rank Correlation on Baseline Simulations

Sensitivity analysis was applied to the risk model to assess the impact of uncertain and highly variable input parameters that were inputted as probability distributions on the estimated number of CPE introductions ($N_{intro_{s}}).$ Spearman rank correlation was used to analyze the impact of these input parameters. Only input parameters with a correlation coefficient >|0.1| with $N_{intro_{s}}$ were included in the result.

#### 4.3.2. One-at-a-Time Sensitivity Analysis

In an additional one-at-a-time (OAT) sensitivity analysis, the most input parameters (non-inferred) (Table 4) were either decreased or increased by 50%. The result of each input adjustment was compared to the baseline result to determine which parameter had the most effect on the expected number of colonized farms. Results were calculated per CPE source (imported livestock, livestock feed, companion animals, hospital patients, and returning travelers). To analyze the effect of changes in input parameters on the ranking of sources for the expected number of farms with CPE introduction, outcomes of each input adjustment were compared to the outcomes of all other input adjustments, including the baseline model, and the frequency of changes in the ranking were counted.

### 4.4. What-If Analysis

Three what-if scenarios were analyzed for their impact on the estimated number of CPE introductions ($N_{intro_{s}})$. The first scenario simulated the effect of less sanitary measures in livestock feed production by increasing the bacteria number in feed ($Ecoli_{concF}$) to the maximum limit for rejecting feed according to GMP+. The second scenario modeled the effect of banning livestock importation from EU member states with insufficient CPE surveillance. In the calculations for this scenario, livestock imports from countries that sampled less than 100 animals for CPE surveillance were excluded from the model calculations. The third scenario evaluated weak compliance with biosecurity protocols on farms. This affected both the risk of introduction from humans and companion animals. The lower biosecurity was mimicked by assuming farm workers did not wash their hands after toilet use, resulting in a higher number of CPE on their hands, and by adjusting the proportion of time a companion animal was present in the animal area $P_{barnC}$. This parameter was set to 0.1 in broiler and pig farms and 0.3 in veal calf farms. All other input parameters were kept at their baseline values in the what-if scenarios.

## 5. Conclusions

Feed and imported livestock are expected to pose the highest risk of CPE introduction to pig, broiler, and veal calf farms. Our risk assessment shows that CPE surveillance should focus on broiler and fattening pig farms, given the highest probability of introduction per farm and the highest total number of introductions, respectively. Our model clearly indicates that we currently do not have sufficient information on the CPE presence in sources, i.e., CPE prevalence in humans, animals, and feed, and the CPE concentration in feed, and that this information is essential for the reliability of this risk estimate and for effective risk mitigation. Therefore, the calculated numbers of exposure and introduction cannot be considered as accurate quantitative estimates of the risk. The ranking of farm types for the total number of introductions in each farm type and for the probability of introduction in individual farm types is, however, robust despite the huge uncertainties in input parameters. More surveillance of CPE prevalence in feed and imported animals, especially veal calves, is essential to improve the certainty of the risk assessment. Banning livestock importation from countries that put little effort into CPE surveillance could reduce the risk from imported livestock.

## Figures and Tables

**Figure 1 antibiotics-11-00281-f001:**
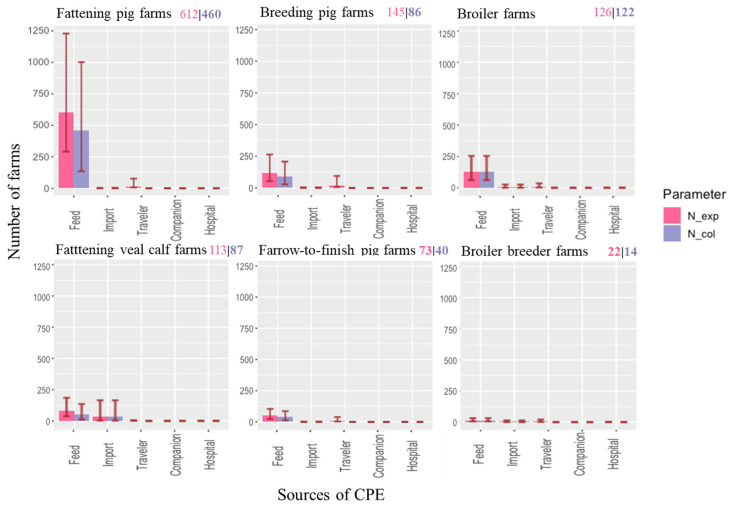
Baseline result: median (whisker: 5th and 95th percentiles) annual number of farms exposed to (red) and colonized by (blue) *CPE* in each farm type from five sources (feed, imported livestock, returning travelers, companion animals, and hospital patients). The color-coded numbers in the right upper corner of each plot are the total number of farms exposed to *CPE* and the total number of farms in which CPE has been introduced.

**Figure 2 antibiotics-11-00281-f002:**
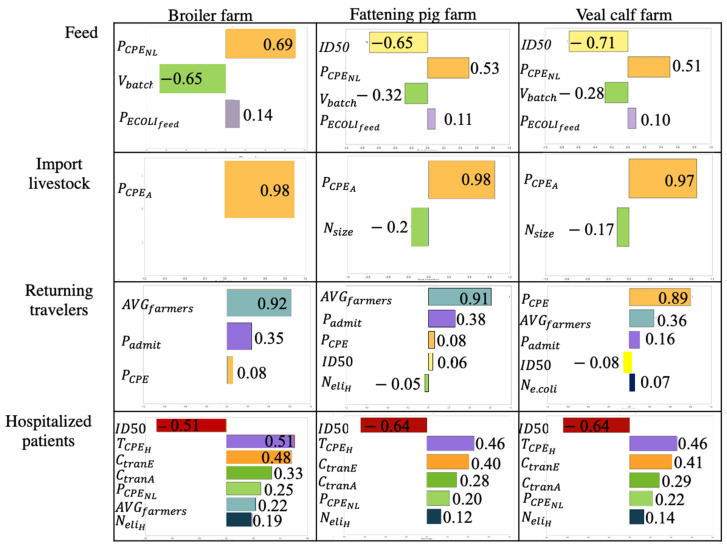
Results of Spearman rank correlation for broiler farm, fattening pig farm, and veal calf farm. Each row shows rank correlation of input parameters with the expected number of *CPE* colonizations from feed, imported livestock, returning travelers, and hospitalized patients. Only input parameters with a Spearman rank correlation coefficient >|0.1| are included in the plots. Spearman rank correlation of companion animals is excluded from the figure because the introduction is zero.

**Figure 3 antibiotics-11-00281-f003:**
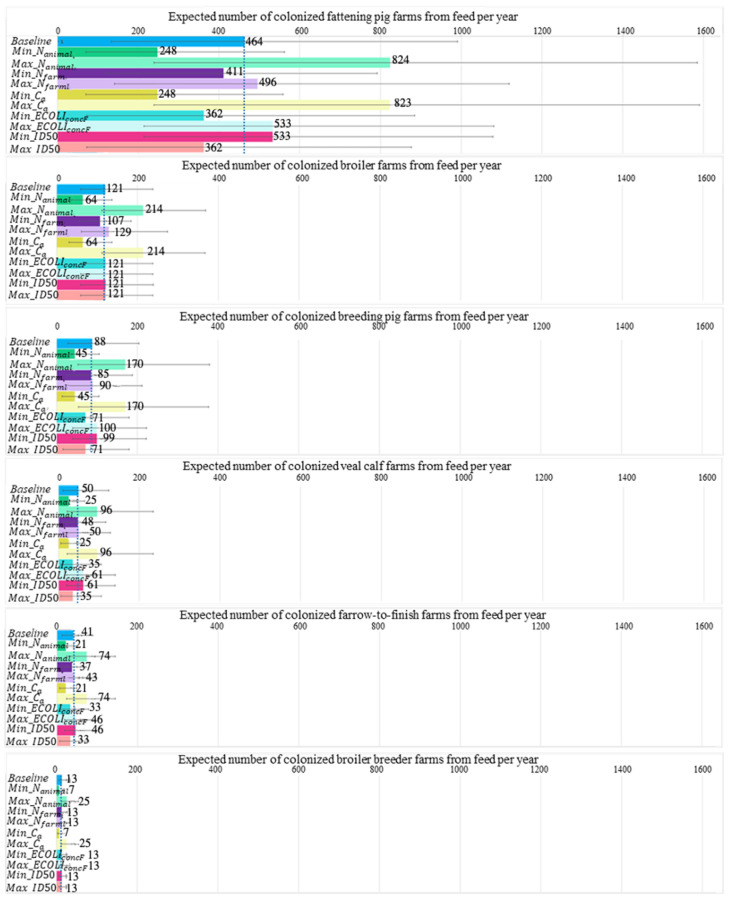
One-at-a-time sensitivity analysis of the number of introductions from feed to six farm types calculated in which one parameter either increases or decreases two-fold. Farm types are ordered according to the highest to lowest number of introductions in the baseline model. Dotted blue line indicates the estimated number of introductions in the baseline model. Only parameters that differed between farm types are included in this figure.

**Figure 4 antibiotics-11-00281-f004:**
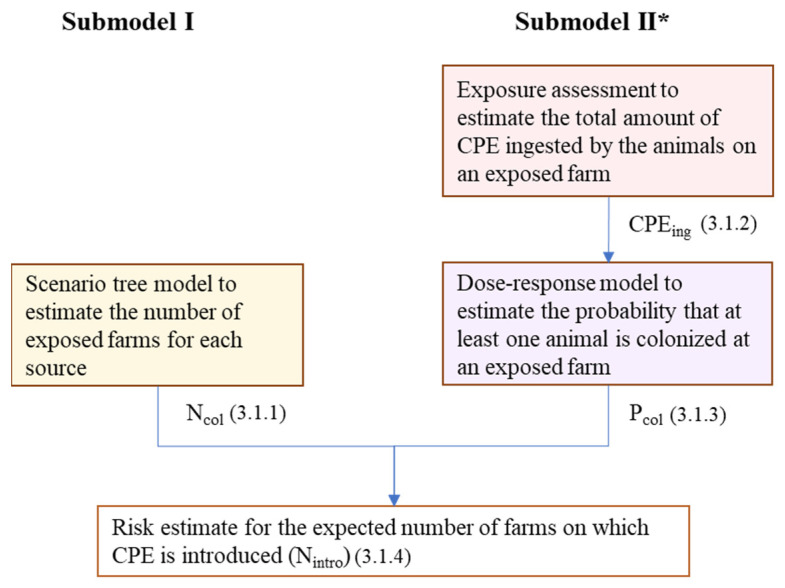
Outline of the risk model to estimate the introduction risk of CPE into Dutch livestock farms from five sources: imported livestock, livestock feed, companion animals (cats and dogs), hospital patients, and returning travelers. * Submodel II is not used for imported livestock because the introduction of a colonized animal into a livestock farm automatically results in colonization of the farm.

**Figure 5 antibiotics-11-00281-f005:**
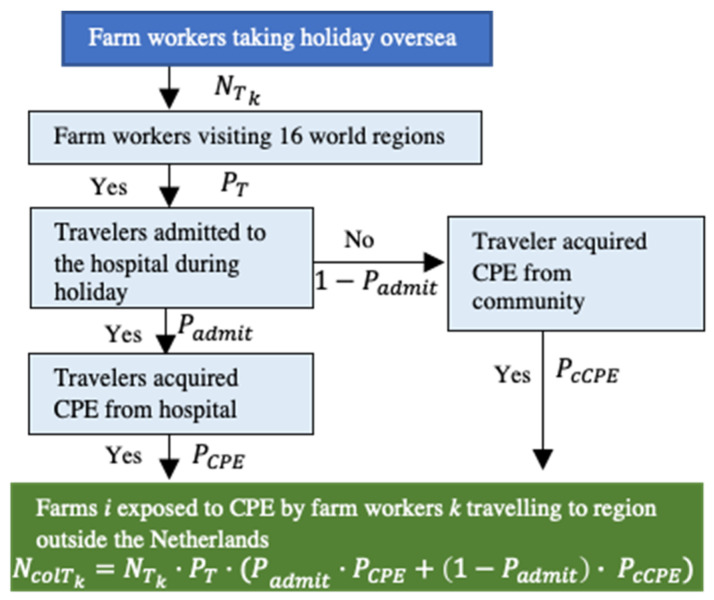
Scenario tree to calculate the number of farms exposed to *CPE* by farm workers returning from travel abroad.

**Table 1 antibiotics-11-00281-t001:** Probability of at least one animal colonized on a farm given exposure of the farm to *CPE.* The companion animal source resulted in zero probability, and there was no calculation for imported livestock.

Farms at Risk	Median Probability of at Least One Animal Being Colonized Given Exposure by a Specific CPE Source (5th and 95th Percentiles)		
Farm Types	Feed	Farm Workers Returning from Travel and Hospital	
		Farm Workers	Veterinarians
Broiler	1.00 (1.00, 1.00)	1 × 10^−4^ (1 × 10^−5^, 8 × 10^−4^)	2 × 10^−6^ (2 × 10^−7^, 2 × 10^−5^)
Broiler breeder	1.00 (1.00, 1.00)	1 × 10^−4^ (1 × 10^−5^, 8 × 10^−4^)	2 × 10^−6^ (2 × 10^−7^, 2 × 10^−5^)
Fattening pig	0.88 (0.22, 1.00)	2 × 10^−7^ (1 × 10^−8^, 5 × 10^−6^)	4 × 10^−9^ (2 × 10^−10^, 9 × 10^−8^)
Breeding pig	0.92 (0.26, 1.00)	2 × 10^−7^ (1 × 10^−8^, 5 × 10^−6^)	4 × 10^−9^ (2 × 10^−10^, 9 × 10^−8^)
Farrow-to-finish	0.92 (0.26, 1.00)	2 × 10^−7^ (1 × 10^−8^, 5 × 10^−6^)	4 × 10^−9^ (2 × 10^−10^, 9 × 10^−8^)
Veal calf	0.73 (0.15, 1.00)	2 × 10^−7^ (1 × 10^−8^, 5 × 10^−6^)	4 × 10^−9^ (2 × 10^−10^, 9 × 10^−8^)

**Table 2 antibiotics-11-00281-t002:** Expected number of farms exposed and colonized combined with the total number of farms to calculate the probability of exposure and colonization for an individual farm of a specific type.

			Broiler	Fattening Pig	Farrow-to-Finish	Veal Calf	Broiler Breeder	Breeding Pig	Total
	Total number of farms in The Netherlands		524	2652	260	1298	255	1601	6590
Expected number	Farms exposed		126	612	73	113	22	145	1091
	Farms colonized		122	460	40	87	14	86	810
Probability per individual farm	Exposure		0.24	0.23	0.28	0.09	0.09	0.09	0.17
	Colonization		0.23	0.17	0.16	0.07	0.05	0.05	0.13
	Probability of exposure due to	Feed	0.229	0.228	0.196	0.059	0.051	0.067	0.148
		Imported livestock	0.004	3 × 10^−4^	0.002	0.025	0.004	0.001	0.007
		Returning traveler	0.008	0.006	0.040	0.006	0.015	0.069	0.143
		Companion animal	0.001	0.004	3 × 10^−4^	0.002	3 × 10^−4^	0.002	0.009
		Hospital patient	1.8 × 10^−4^	0.001	2 × 10^−4^	4 × 10^−4^	8 × 10^−5^	5 × 10^−4^	0.003

**Table 3 antibiotics-11-00281-t003:** What-if analysis related to probability of colonization in feed, restriction on import of animals from countries with weak surveillance for *CPE*, and less strict biosecurity practice in local farms.

Scenario	CPE Source Affected	Parameter Changed	Baseline Number of Introductions from Affected Source (95% Range)	Changed Number of Introductions from Affected Source (95% Range)
Contamination of *E. coli* in feed reaches concentration of maximum rejection limit according to GMP+	Feed	$Ecoli_{concF}$	767 (244, 1679)	775 (246, 1668)
The Netherlands only allows import of livestock from EU member states that sample ≥100 animals in CPE surveillance	Imported livestock	$P_{CPE_{A}}$	48 (4, 214)	14 (0, 58)
Lower biosecurity: companion animals have full access to livestock areas in broiler, pig, and veal calf farms	Companion animals	$P_{barnC}{}_{}$	0 (0, 0)	2 (1, 7)
Lower biosecurity: non-compliance with hand hygiene	Travelers and hospitalized patients	$Ecoli_{hand}$	1 × 10^−4^ (9 × 10^−6^, 8 × 10^−4^)	4 × 10^−3^ (3 × 10^−4^, 3 × 10^−2^)

## Data Availability

Data retrieved from publicly available sources are provided in the references. All other data are provided in the Appendix A.

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
