# Peer review of "Quantitative Risk Assessment for the Introduction of Carbapenem-Resistant Enterobacteriaceae (CPE) into Dutch Livestock Farms"

_antibiotics, 2022, doi:10.3390/antibiotics11020281_

Round 1

Reviewer 1 Report

The authors presented a very interesting paper about an important issue. Indeed, the introduction of CPE (and other multidrug-resistant organisms) into livestock farms contributes to the risk of human contamination.

The present study allowed to discriminate the most important risk factors for CPE introduction into livestock farms.

The results are clearly presented, the discussion is interesting and the methods are well described and seem relevant.

Therefore, I think that this paper is suitable for publication in Antibiotics.

I just have several questions

  1. The authors indicated that according to the EU surveillance protocol, member states must collect samples for searching CPE. In the current EU surveillance protocol, what samples are performed? dungs? Soil in barns? …
  2. Farms included in the study seem to have a single livestock activity (fattening pig farms, broiler breeder farms…). Did certain farms have more diverse livestock activities? In this case, do the authors consider that multiple livestock in the same farm could potentially represent a risk of CPE?
  3. I do not totally agree with the choice of authors of excluding wastewater from hospitals. Indeed, even though CPE are effectively removed in the wastewater treatment facilities, several studies demonstrated the presence of resistance genes in water downstream these facilities. Therefore, was it not possible for livestock animals to be contaminated by resistance genes, which could be incorporated in their digestive microbiota’s Enterobacteriaceae?

Author Response

Reviewer 1: I just have several questions

1. The authors indicated that according to the EU surveillance protocol, member states must collect samples for searching CPE. In the current EU surveillance protocol, what samples are performed? dungs? Soil in barns? …

Response : Current EU surveillance take cecal samples resistant bacteria from fattening pig, broiler, and veal calf at slaughter house.

Actions: Added in line 55-57: “Cecal samples from live fattening pig, veal calf, and broiler at slaughterhouse were collected and tested for resistant genes.”

2. Farms included in the study seem to have a single livestock activity (fattening pig farms, broiler breeder farms…). Did certain farms have more diverse livestock activities? In this case, do the authors consider that multiple livestock in the same farm could potentially represent a risk of CPE?

Response: We checked the numbers of companies with a combination of livestock production. Only a small proportion of broiler and parent broiler farms have combination of livestock species (less than 10 percent of total broiler and parent broiler farms). In pig producing farms, 15 % of fattening pig farms have multiple livestock species in the same farms. As this reviewer correctly indicates, these combination farms may be at higher risk, as they have an increased number of sources because of the combination. Nevertheless, the size of the separate parts of the farm is almost always lower than in specialized farms. Because feed is the main driver of the risk the smaller size will reduce the risk of these combined farms. For that reason, and because of the limited proportion of combined farms we did not consider these diverse farms. These farms, might, however have a risk of CPE introduction if e.g. less strict biosecurity is applied. We did not have data to estimate this.

Actions: add-on in line 484-485:

“Mixed species livestock farms were not considered in the risk assessment because they represented a small proportion of local farms 43

3. I do not totally agree with the choice of authors of excluding wastewater from hospitals. Indeed, even though CPE are effectively removed in the wastewater treatment facilities, several studies demonstrated the presence of resistance genes in water downstream these facilities. Therefore, was it not possible for livestock animals to be contaminated by resistance genes, which could be incorporated in their digestive microbiota’s Enterobacteriaceae?

Response: We agree that wastewater is a potential source of CPE. However, we decided to exclude this source because in the Netherlands the livestock of our concern (veal calf, fattening pig, breeding pig, broiler, and broiler parents)  is predominantly raised in closed barns where the animals drink tap water or groundwater from a well. Should we have included grazing animals, such as dairy cattle and sheep, natural water sources would need to be included as a potential source. 

Action: We clarified this further in line 447-451.

“And although small traces of CPE could be present in surface water due to overflow from rainfall, the vast majority of the meat-producing animals of our concern (veal calf, fattening pig, breeding pig, broiler, and broiler breeder) were raised in closed system where they drink tap water. This water source undergoes extensive purification, ensuring no traces of resistant bacte-ria such as CPE 35-37.”

Reviewer 2 Report

In this study, authors assumed that CPE occurrence in food-producing animals could be underestimated, as sampling for their detection is limited in EU, and therefore, they tried to identify potential risk factors for CPE introduction in different types of farms. By reviewing literature, hospital patients, returning travelers from abroad, companion animals, wild animals, wastewater from hospitals, imported livestock and animal feed were recognized as potential CPE sources. Then, authors justified why they excluded the parameters of wastewater from hospitals as well as the presence of wild mammals, birds and rodents from the model. They also considered that companion animals as a potential CPE source (lines 66-89). In addition, it was mentioned that if strict biosecurity measures were applied on farms, colonized companion animals did not pose a threat for the exposure of farm animals (line 356). Why didn’t they exclude companion animals from the beginning as they did with rodents? I am wondering, when strict biosecurity measures are followed, is it more possible for farm animals to contact with companion animals or rodents? I believe the latter.

Thirteen percent of Dutch farms were estimated to be colonized by CPE each year, mainly via feed (line 273) though CPE-producing bacteria have not been identified in feed. I believe that model yielded too many overestimations due to the lack of available data, and thus results could be misleading.

Author Response

Reviewer 2:

In this study, authors assumed that CPE occurrence in food-producing animals could be underestimated, as sampling for their detection is limited in EU, and therefore, they tried to identify potential risk factors for CPE introduction in different types of farms. By reviewing literature, hospital patients, returning travelers from abroad, companion animals, wild animals, wastewater from hospitals, imported livestock and animal feed were recognized as potential CPE sources. Then, authors justified why they excluded the parameters of wastewater from hospitals as well as the presence of wild mammals, birds and rodents from the model. They also considered that companion animals as a potential CPE source (lines 66-89). In addition, it was mentioned that if strict biosecurity measures were applied on farms, colonized companion animals did not pose a threat for the exposure of farm animals (line 356). Why didn’t they exclude companion animals from the beginning as they did with rodents? I am wondering, when strict biosecurity measures are followed, is it more possible for farm animals to contact with companion animals or rodents? I believe the latter.

Response: Companion animals were included as they were considered to have a higher probability of being colonized with CPE, since CPE has been detected in local companion animals in recent years (as stated in lines 681-685) and because a considerable number of companion animals is imported from abroad. Rodents only move locally, so no CPE from abroad is expected there. This is explained in lines 451-455.

Thirteen percent of Dutch farms were estimated to be colonized by CPE each year, mainly via feed (line 273) though CPE-producing bacteria have not been identified in feed. I believe that model yielded too many overestimations due to the lack of available data, and thus results could be misleading.

Response: We agreed that 13 percent is likely to be an overestimation. However, resistant bacteria and E. coli were never included in the feed surveillance in the Netherlands and in most European Union Member states, thus, we cannot exclude this possibility. Furthermore, as indicated in the manuscript (line 87), for the purpose of AMR surveillance, in particular the ranking of the risk of the CPE sources and the farm types is important and the sensitivity analysis showed that to be robust. 

In line 325 we highlighted the uncertainty surrounding the feed. We also explained the weaknesses of the current CPE surveillance in animal populations and the lack of CPE surveillance in feed. The latter also excludes detection of CPE in feed.

“According to our model, thirteen percent of Dutch farms are estimated to be colonized by CPE each year, mainly via feed, which is clearly an overestimation as such a percentage of farms being colonized is detectable under the current national surveillance protocol15, 16. Still undetected CPE presence in Dutch livestock is possible, as the current national surveillance protocol was designed to detect at least one colonized animal with 95% certainty if prevalence would be 1%15. However, this surveillance protocol does not take into account clustering of colonization at farm level, which decreases the sensitivity of the surveillance. Furthermore, introductions could have escaped detection, because most farms for meat production (broiler, fattening pig, and veal calf) apply an all-in-all-out system that produces more than one batch of livestock annually while the national surveillance collects samples only once a year from a single animal per batch at slaughter from part of the farms. Thus, for each farm unit multiple samples distributed over time are necessary to calculate an accurate prevalence17. “

Action: To prevent misleading result, we add a paragraph to line 418-425

The outcome of this introduction risk assessment is used to rank farm types and sources for their CPE introduction risk. The results for the absolute numbers of exposures and introductions have a large uncertainty and cannot be viewed as accurate quantitative risk estimates. Results of the sensitivity analysis give good indications of the uncertain input parameters that have the largest impact on model results. Parameters with both large uncertainty and large impact are important knowledge gaps that can be targeted in future studies. Despite these uncertainties, ranking of farm types and sources was robust and the outcome of this risk assessment can thus be used for targeted CPE surveillance33-35.”

Reviewer 3 Report

Natcha Dankittipong and colleagues (antibiotics-1575401) presented a risk assessment of the type of sources for Carbapenem-resistant Enterobacteriaceae (CPE) into Dutch livestock farms, however, it is hard to understand the relationship between overall prevalence and introduction. Typical meta-analysis conducted meta-prevalence for global (doi: 10.3390/microorganisms7100461.), or national-wide (doi: 10.3389/fvets.2020.00521.). Did the math model for such analysis conducted before? any previous publications to support what has been conducted?  Most of the result sections are very hard to follow.

The references for data acquisition are also needed, please make them supplemental documents.

the way for conducting the meta-data collection should be included in the methods.

Line 79-81, please provide the citation for these statements.

Line 79-89, the rational study part should move to material and method.

Table 2 should keep within one page, and what does "Prob" mean?

The information is the bracket is not easy to follow, ??????, ????????, 

?????ℎ...... please make a table to clarify this.

There are numerous jargon across the manuscript, i.e.

Sensitivity analysis-Spearman rank correlation, One-at-a-time sensitivity analysis, What-if analysis, should be clarified before introduction in the results. What are the purposes of those analyses? Accordingly, figure 2 is very hard to understand and interpret.

There are too many subscripts, which are spreading around the whole manuscript, and it is very hard to differentiate and follow individual types and make comparisons. Same for figure 4.

The aim for table 3 is also not clear, why it is needed.

There are errors in the references, and many new and relevant references should be included.

The way for results delivered should be adjusted, use the aim first, then how you conducted and what is the most significant results.

Finally, the quality of all figures should be improved.

Author Response

Reviewer 3:

Natcha Dankittipong and colleagues (antibiotics-1575401) presented a risk assessment of the type of sources for Carbapenem-resistant Enterobacteriaceae (CPE) into Dutch livestock farms, however, it is hard to understand the relationship between overall prevalence and introduction. Typical meta-analysis conducted meta-prevalence for global (doi: 10.3390/microorganisms7100461.), or national-wide (doi: 10.3389/fvets.2020.00521.). Did the math model for such analysis conducted before? any previous publications to support what has been conducted? 

Response: For the exposure assessment, we followed the quantitative risk assessment protocol “Import Risk Analysis for Animals and Animal Products” from The World Organisation for Animal Health (OIE), using the approach of the risk assessment for Classical Swine Fever Virus introduction to European Member States (De Vos et al, 2004). To assess the probability of infection upon exposure, the microbial risk assessment protocol as given by the Codex Alimentarius (FAO) was followed. A risk assessment model of resistant bacteria introduction to livestock was not conducted before. However, qualitative assessment and case detection of Carbapenemase resistant bacteria in livestock has been illustrated by European Food Safety Authority since 2013 (EFSA, 2013).

Estimates of non-pathogenic AMR prevalence (such as resistant bacteria) are hard to find, and indeed this is one of the main uncertain parameters of this study. We relied on inference of the prevalence in different sources from prevalence estimates of surveillance or proxy measures such as the prevalence in humans. This uncertainty is addressed in detail in the sensitivity analysis. A key outcome of this study is that feed could be an important risk of introduction if indeed CPE is present in feed. We advise further investigation into this source given the lack of data on CPE prevalence and concentration in feed. 

Action: We add references to De Vos et al 2004, Codex Alimentarius, and Haas et al, 2014  in line 432-435; “This quantitative risk assessment followed the guidelines for import risk assessment given by the World Organisation for Animal Health (OIE) 35, 36 to assess the risk of exposure of farms and the guidelines for microbial risk assessment given by the Codex Alimentarius to assess the risk of infection upon exposure37, 38

Most of the result sections are very hard to follow. The way for results delivered should be adjusted, use the aim first, then how you conducted and what is the most significant results.

Response: In order to understand the most important output of the risk assessment (i.e. the number of introductions), we must explain the underlying risk estimates which are the number of exposures and the probability of colonization. Thus, we would like to keep the structure of exposure, colonization, and introduction. To guide the reader, we have included a brief description of the structure of the risk assessment at the start of the result section.

Action: We added a description of the line of the risk assessment in lines 81-91

“To estimate the risk of introduction, first the number of farms exposed to CPE sources (section 2.1) and the probability of colonization after exposure (section 2.2) are estimated. These are combined into the risk of introduction by calculating the number of expected introductions (section 2.3). The sensitivity of model output is determined by two methods of sensitivity analysis (section 2.4). First, spearman correlation coefficients were used to identify important uncertain parameters. Second, one-at-a-time sensitivity was used to investigate the robustness of the ranking of risks to changes in each of the input parameters. Finally, different scenarios with respect to contamination of feed, restrictions on imports, and biosecurity were studied (section 2.5).

The references for data acquisition are also needed, please make them supplemental documents. the way for conducting the meta-data collection should be included in the methods.

Response: For the data that was obtained from public repositories and databases, we have added the URL of imported animals retrieved from RVO and the issue number of the report on CPE prevalence in the European Union (livestock and human)  by the European Food Safety Authority in the references. The number of imported animals from the CBS database needs a specific query to retrieve it, which has been added to the supplementary information.

Action: In supplementary S6.1 the query for cbs.nl was added.

Line 79-81, please provide the citation for these statements. Line 79-89, the rational study part should move to material and method.

Action: two more references are added in line 451 and we moved the paragraph to line 445-455. 

Table 2 should keep within one page, and what does "Prob" mean?

Response: It means probability of exposure due to CPE sources. The table overlapped two pages which deleted the full description.

Action: table moved to line 152.

The information is the bracket is not easy to follow, ???????????????????ℎ...... please make a table to clarify this. #add a table of parameters

Action: these parameters and descriptions were added in table 5.

There are numerous jargon across the manuscript, i.e. Sensitivity analysis-Spearman rank correlation, One-at-a-time sensitivity analysis, What-if analysis, should be clarified before introduction in the results. What are the purposes of those analyses? Accordingly, figure 2 is very hard to understand and interpret.

Response: Sensitivity analyses, such as performed in this manuscript, are standard methods in risk assessment. We understand that these are not always known and added descriptions of the methods at the start of the results section for sensitivity analysis in line 154 and for the what-if analysis in line 276. This should also make figure 2 clearer to understand.

Action: add-ons in line 154-163:

“First Spearman rank correlation, a non-parametric metric between -1 and 1, was calculated for all input parameters with an uncertainty distribution to estimate the extent to which these input parameters determined the model results for each source (section 2.4.1). Secondly, one-at-a-time (OAT) sensitivity analysis was performed (section 2.4.2). In this additional sensitivity analysis, the value of a single input parameter was either increased or decreased. The outcome of each adjustment was compared to the baseline scenario to investigate the impact of all input parameters on the estimated number of introductions. The OAT sensitivity analysis was performed separately for each source. Then, to evaluate if changes in input parameters would affect the ranking of sources, we compared the results of the OAT sensitivity analysis across sources (section 2.4.3).”

line 276-279:  “The effect of higher contamination levels in feed, less strict biosecurity at farm level, and a ban on livestock imports from countries sampling less than 100 animals for CPE surveillance were explored by adjusting input parameters and evaluating the model outcome (number of introductions) in what-if scenario analysis.  “

The aim for table 3 is also not clear, why it is needed.

We wanted to show the robustness of the ranking. However, we agree now with the reviewer that this table does not add much and we have therefore removed it to Supplementary Information. We added line 252-259 to clarify how the OAT sensitivity analysis was used to check the robustness of the top ranking of the feed source.

There are too many subscripts, which are spreading around the whole manuscript, and it is very hard to differentiate and follow individual types and make comparisons. Same for figure 4.

Action: We reduced the heavy subscriptions that indicated farm types (i), European member states (j) and world regions (m) throughout the manuscript

There are errors in the references, and many new and relevant references should be included.

Response: we included more references and corrected the references to data

Finally, the quality of all figures should be improve

We will adjust this according to the standards of the journal and attached as additional file.

Reviewer 4 Report

This important and  impressively detailed and illustrated paper documents significant public health concerns about the development of antibiotic-resistant strains of Enterobacteriaceae/ E. Coli in Dutch livestock farms. These findings should apply equally to the rest of  Europe ass well as globally.  The authors have identified the problem, summarized the findings, and made practical suggestions to help mitigate the societal impact.  

Minor Comments  -- The general excellence of the paper had a few early awkward phrasings --Suggest rewording - as follows: Abstract, Lines 15-16. Suggest "Early detection of emerging of Carbapenum  ------- is essential to control spread of CPE." 

Line 35. Suggest " ---challenge between development of  resistant ---- and treatment needs." 

Line 36.  "--- the speed that resistance emerges  are --- ".

Line 47. Suggest " --enables development of CPE cases--- ".

Line 190.  "pig farms"   Line 347. "-- but the risk of these sources were was found --- ".

Author Response

Reviewer 4:

This important and  impressively detailed and illustrated paper documents significant public health concerns about the development of antibiotic-resistant strains of Enterobacteriaceae/ E. Coli in Dutch livestock farms. These findings should apply equally to the rest of  Europe ass well as globally.  The authors have identified the problem, summarized the findings, and made practical suggestions to help mitigate the societal impact.  

Minor Comments  -- The general excellence of the paper had a few early awkward phrasings --Suggest rewording - as follows: Abstract, Lines 15-16. Suggest "Early detection of emerging of Carbapenum  ------- is essential to control spread of CPE." 

Line 35. Suggest " ---challenge between development of  resistant ---- and treatment needs." 

Line 36.  "--- the speed that resistance emerges  are --- ".

Action : we edited lines 15-16, 35 and 36 based on the suggestions given by the reviewer

Line 47. Suggest " --enables development of CPE cases--- ".

Action : add-on in line 47

Line 190.  "pig farms"   

Action : edited in line 236

Line 347. "-- but the risk of these sources were was found --- ".

Action : add-on in line 399

Round 2

Reviewer 2 Report

I believe that this version of the manuscript is improved since authors added a last paragraph in discussion explaining the restrictions of their model results. In my opinion the word 'Quantitative'  should be omitted from the title of the paper or the title should be rephrased. 

Reviewer 3 Report

This manuscript has been made some improvement, I would suggest to add most recent literature:

10.3390/antibiotics11020236

10.3390/antibiotics9040186 

10.3390/antibiotics10020177 

10.3390/microorganisms7100461